# Prevention of Compression Fracture in Osteoporosis Patients under Minimally Invasive Trans-Foraminal Lumbar Interbody Fusion with Assistance of Bone-Mounted Robotic System in Two-Level Degenerative Lumbar Disease

**DOI:** 10.3390/medicina58050688

**Published:** 2022-05-23

**Authors:** Hui-Yuan Su, Huey-Jiun Ko, Yu-Feng Su, Ann-Shung Lieu, Chih-Lung Lin, Chih-Hui Chang, Tai-Hsin Tsai, Cheng-Yu Tsai

**Affiliations:** 1Division of Neurosurgery, Department of Surgery, Kaohsiung Medical University Hospital, Kaohsiung Medical University, Kaohsiung 807, Taiwan; latimeria8@hotmail.com (H.-Y.S.); suyufeng2000@yahoo.com.tw (Y.-F.S.); e791125@gmail.com (A.-S.L.); chihlung1@yahoo.com (C.-L.L.); chchang20@gmail.com (C.-H.C.); teishin8@hotmail.com (T.-H.T.); 2Department of Surgery, Kaohsiung Medical University Hospital, Kaohsiung Medical University, Kaohsiung 807, Taiwan; 3Graduate Institute of Medicine, College of Medicine, Kaohsiung Medical University, Kaohsiung 807, Taiwan; o870391@yahoo.com.tw; 4Department of Biochemistry, College of Medicine, Kaohsiung Medical University, Kaohsiung 807, Taiwan; 5Graduate Institute of Clinical Medicine, College of Medicine, Kaohsiung Medical University, Kaohsiung 807, Taiwan; 6Faculty of Medicine, College of Medicine, Kaohsiung Medical University, Kaohsiung 807, Taiwan; 7Ph.D. Program in Environmental and Occupational Medicine, College of Medicine, Kaohsiung Medical University and National Health Research Institutes, Kaohsiung 807, Taiwan; 8Post Baccalaureate Medicine, College of Medicine, Kaohsiung Medical University, Kaohsiung 807, Taiwan

**Keywords:** osteoporosis, robotic surgery, screw-loosening rate, spinal fixation, vertebral compression fracture

## Abstract

*Background and Objectives:* Minimally invasive spine surgery reduces destruction of the paraspinal musculature and improves spinal stability. Nevertheless, screw loosening remains a challenging issue in osteoporosis patients receiving spinal fixation and fusion surgery. Moreover, adjacent vertebral compression fracture is a major complication, particularly in patients with osteoporosis. We assessed long-term imaging results to investigate the outcomes of osteoporosis patients with two-level degenerative spine disease receiving minimally invasive surgery with the assistance of a robotic system. *Materials and Methods:* We retrospectively analyzed consecutive osteoporosis patients who underwent minimally invasive surgery with the assistance of a robotic system at our institution during 2013–2016. All patients were diagnosed with osteoporosis according to the World Health Organization criteria. All patients were diagnosed with two levels of spinal degenerative disease, including L34, L45, or L5S1. The study endpoints included screw-loosening condition, cage fusion, and vertebral body heights of the adjacent, first fixation segment, and second fixation segments before and after surgery, including the anterior, middle, and posterior third parts of the vertebral body. Differences in vertebral body heights before and after surgery were evaluated using the F-test. *Results:* Nineteen consecutive osteoporosis patients were analyzed. A lower rate of screw loosening was observed in osteoporosis patients in our study. There were no significant differences between the preoperative and postoperative vertebral body heights, including adjacent and fixation segments. *Conclusions:* According to our retrospective study, we report that minimally invasive surgery with the assistance of a robotic system provided better screw fixation, a lower rate of screw loosening, and a lesser extent of vertebral compression fracture after spinal fixation and fusion surgery in osteoporosis patients.

## 1. Introduction

Transpedicular screw fixation is a popular technique in spinal fusion surgery. With the global increase in the proportion of elderly individuals, the necessity for spinal surgery for osteoporosis patients is frequently encountered. Owing to the fragile characteristics of the elderly, complications, including screw-loosening events, pseudarthroses, and adjacent kyphosis resulting from compression fractures, may occur after spinal surgery in osteoporosis patients [1]. In spinal fixation and fusion surgery, screw-loosening events continue to remain a major complication [2], and cage subsidence is relevant to the severity of osteoporosis [3]. Recently, adjacent vertebral compression fractures have attracted attention after spinal fixation and fusion in osteoporosis patients [4]. Less fatty degeneration in the paraspinal muscle has been noted in patients undergoing minimally invasive surgery than in those undergoing conventional open surgery [5]. Although minimally invasive procedures have led to improvements in the surgical outcomes of lumbar interbody fusion, the most recent rate of nonunion after interbody fusion still ranged from 7% to 20%, with a higher incidence in osteoporosis patients due to repeated puncture and imperfect screw location [6]. Many studies concerning robotic spine surgery have discussed the accuracy of screw placement, exposure to radiation, and learning curve; however, the clinical outcomes of minimally invasive surgery with the assistance of robotic systems in osteoporosis patients have rarely been discussed [7].

In this study, we aimed to investigate the clinical outcomes of minimally invasive surgery with the assistance of a robotic system by evaluating long-term imaging results for osteoporosis patients with two-level degenerative spine disease who underwent this treatment.

## 2. Materials and Methods

### 2.1. Study Patients

This study was approved by the Institutional Review Board/Ethics Committee of Kaohsiung Medical University Hospital, Kaohsiung, Taiwan (KMUH103-3T15), and informed consent was obtained from all patients. All methods described in this study were performed in accordance with the relevant guidelines and regulations of the Declaration of Helsinki and an institutional review board.

We retrospectively reviewed the medical records of all patients who were diagnosed with two-level spinal degenerative disease, e.g., lumbar spondylolisthesis, and underwent two-level minimally invasive surgery assisted with a robotic system, including levels L34, L45, or L5S1, during 2013–2016. The flow chart of our study is shown in Figure 1.

### 2.2. Inclusion and Exclusion Criteria

The inclusion criteria were: (1) diagnosis of osteoporosis according to the World Health Organization criteria for osteoporosis, with a T-score of less than −2.5 standard deviations (SD) confirmed by dual-energy X-ray absorptiometry; and (2) no in situ or adjacent vertebral compression fracture on lumbar plain film and computed tomography. The exclusion criteria were: (1) malignancy-related spinal lesions, (2) a T-score of more than −2.5 SD, or (3) a vertebral compression fracture on the lumbar plain film and computed tomography.

After minimally invasive surgery assisted with the robotic system, all patients were given the same postoperative care, including wound care, bedrest time, and rehabilitation programs.

### 2.3. Radiological Assessment

The study endpoints included: (1) screw-loosening condition, assessed through radiography or computed tomography; (2) cage fusion, assessed through radiography or computed tomography; and (3) vertebral body heights of adjacent, first fixation, and second fixation segments, before and after surgery, including the anterior, middle, and posterior third parts of the vertebral body. All basic clinical data for the patients, including age, sex, body mass index, T-score, bone mineral density, surgical level, and duration of follow-up, are shown in Table 1.

### 2.4. Surgical Technique

Before the operation, the Renaissance robotic system was used to plan an adequate tract for screw placement. During the operation, the patient was placed in the prone position and stabilized under general anesthesia. The mounting equipment of the Renaissance robotic system was applied on the back, then registration with anterior–posterior and oblique plain films was performed. Subsequently, robot assembly and drilling execution were performed. A Kirschner wire was inserted along the tapped canal guided by the robotic system (Figure 2).

A 2 cm incision was made along the line between two pins inserted into the upper and lower pedicles on one side. Then, a tubular dilator was inserted in the direction of the facet joint. Under a microscope, the isthmus, posterior lamina, inferior facet, and ligamentum flavum were resected, followed by placement of the interbody cage after discectomy and preparation of the endplate of the vertebral body. Subsequently, transpedicle screws were inserted along the previously inserted Kirschner wires. The same procedure was performed on the contralateral side.

### 2.5. Statistical Analysis

All data are expressed as means and standard deviations for variables, and differences between preoperative and postoperative vertebral body heights were evaluated using the F-test, with statistical significance set at *p* < 0.05. Statistical analyses were performed using the Statistical Package for the Social Sciences (version 17; IBM, Armonk, NY, USA).

## 3. Results

Nineteen consecutive patients with osteoporosis were included and analyzed. The demographic data and clinical characteristics of patients are shown in Table 1. The ratio of males: females was 3:16. The mean follow-up period was 43 months, mean age was 69 years, mean T-score was −3.3, mean body mass index was 25.6 kg/m^2^, and bone mineral density was 0.8 g/cm^3^, with the screw-loosening rate being 14.47% and cage fusion rate being 89.47%. There were no significant differences in vertebral body heights at the adjacent level, first fixation, or second fixation segments between the preoperative and postoperative measurements (Table 2). The results revealed no significant variation in new-onset vertebral compression fractures in situ or in adjacent segments.

No device-related complications, including broken screws or cage subsidence, occurred, and no other complications, such as wound infection, postoperative pneumonia, visceral injury, and apparent neurologic deficit, were reported.

## 4. Discussion

The present study showed that osteoporosis patients had better screw fixation, a lower rate of screw loosening, and a lesser extent of vertebral compression fracture after spinal fixation and fusion surgery. There were no significant differences between the preoperative and postoperative vertebral body heights and adjacent and fixation segments.

Transpedicle screw fixation is an essential step in spinal fusion surgery; however, it is associated with many complications, including screw-loosening events, pedicle breakdown, and screw pull-out. In the literature, the screw-loosening rate is 1−15% in non-osteoporosis patients, whereas up to 60% screw loosening has been reported in osteoporosis patients. The reasons for the high screw-loosening rate and non-fusion cage rate could be poor fixation environments for fusion. Consequently, screw loosening is a major problem after spinal fixation with screws, especially in osteoporosis patients [8]. Various instruments have been proposed to solve this problem, including conical pedicle screws, screws of increased diameter or length, expandable screws, and cannulated screws for bone cement augmentation [9].

Minimally invasive surgery is frequently applied for degenerative spine disease. Although the paraspinal muscle is protected during minimally invasive surgery, a high screw-loosening rate of up to 22.2% is still reported in osteoporosis patients undergoing minimally invasive transforaminal lumbar interbody fusion surgery [10]. The insertion of transpedicle screws with the assistance of a robotic system is a new technique in spinal surgery, with most of the literature focusing on the accuracy, radiation exposure, and learning curve [7]. However, thus far, reports concerning the benefits of minimally invasive surgery with the assistance of a robotic system in osteoporosis patients are rare.

In this study, the screw-loosening rate of 14.47% was relatively low compared with the rates of 22.2% in patients receiving transforaminal lumbar interbody fusion and 32.3% in patients receiving posterior lumbar interbody fusion in populations with osteoporosis [10,11]. A lower screw-loosening rate in osteoporosis patients was demonstrated in our study. In addition, the spinal fusion rate of 89.47% was comparable to the fusion rate of 89.7% in elderly patients with osteoporosis reported in the literature [12]. Regardless of bone mineral density, a general rate of 4% of adjacent vertebral compression fractures is reported in conventional open spinal surgery, and osteoporosis is regarded as a risk factor for adjacent vertebral compression fracture after spinal fixation and fusion surgery [4]. In our study, there were no significant differences in decreased vertebral body heights in fixation and adjacent segments between preoperative and postoperative evaluations, and there were no new-onset in situ or adjacent vertebral compression fractures in spinal fixation and fusion observed after minimally invasive surgery with the assistance of the robotic system for osteoporosis patients after a long-term follow-up.

Lower screw-loosening rates and fewer incidents of vertebral compression fracture were noted after spinal fixation and fusion through minimally invasive surgery with the assistance of the robotic system in osteoporosis patients in this study. Although less destruction of the paraspinal musculature attributed to minimally invasive surgery has been proven [5], repeated tapping on the pedicles with trocars under guidance of fluorography is still an important process that is necessary during the operation. On the contrary, less destruction of pedicles from omission of tapping with trocars under the assistance of a robotic system can be achieved, and the lower screw-loosening rate in our study might be related to more intact pedicles. The stability of the spine is attributed to the strength of the paraspinal musculature, and fixation of the trans-pedicle screw leads to less vertebral compression fracturing. Finally, our results indicated that minimally invasive spinal surgery with the assistance of bone-mounted robotic system is also an effective treatment to achieve spinal fixation and fusion for osteoporosis patients compared with other methods, such as cement-augmented pedicle screw use or using only applied minimally invasive spinal surgery for osteoporosis patients [13,14].

### Limitation

The limitation of this study was the small sample size; nevertheless, data from a special group of patients with osteoporosis receiving an advanced procedure of minimally invasive surgery with the assistance of a robotic system for solving the complications after spinal instrumentation for fixation and fusion were collected. To overcome this limitation, larger samples from multicenter or larger randomized trials should be considered in the future. In addition, age should also be taken into consideration for further evaluation.

## 5. Conclusions

Osteoporosis-related complications are common in spinal fixation and fusion surgeries. Lower screw-loosening rates and fewer adjacent vertebral compression fractures were noted in this study as compared to conventional spinal surgery. The reduced adjacent vertebral compression fracturing might be attributed to a stable environment and a lower screw-loosening rate due to robotic assistance and less destruction of the paraspinal muscle in minimally invasive surgery.

## Figures and Tables

**Figure 1 medicina-58-00688-f001:**
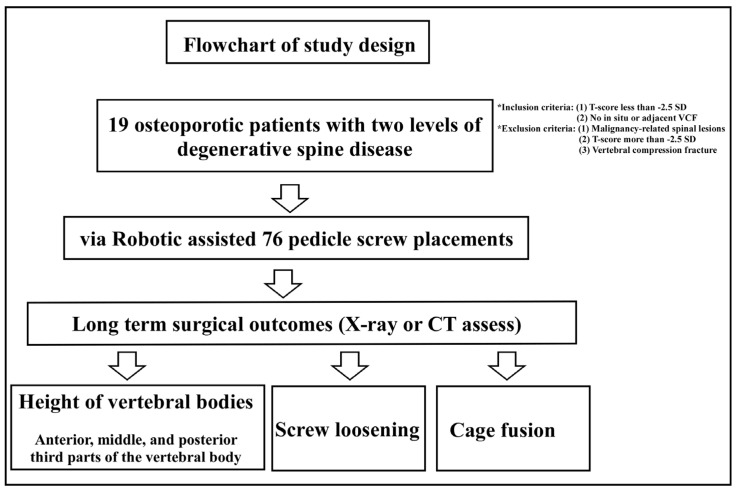
Flowchart of the study. In total, 19 osteoporotic patients diagnosed with two levels of degenerative spine disease were enrolled in this retrospective study and all patients received spinal fixation and fusion surgery over two spinal segments. The above procedure was performed by minimally invasive surgery with the assistance of a robotic system. The patients were diagnosed with osteoporosis according to the World Health Organization’s diagnostic criteria for osteoporosis. After the operation, the surgical outcomes were evaluated, including the heights of vertebral bodies, screw-loosening rate, and spinal fusion rate.

**Figure 2 medicina-58-00688-f002:**
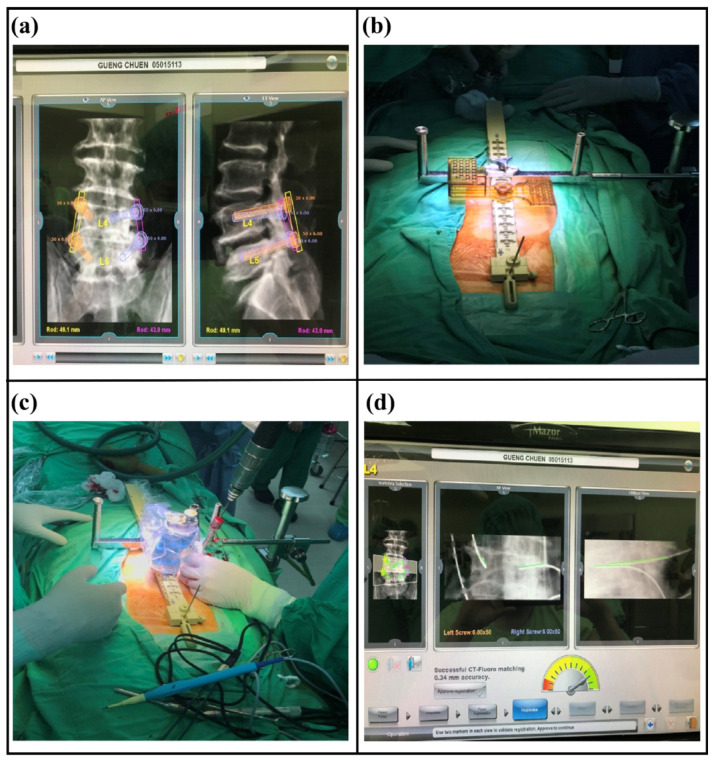
(**a**) Before the operation, the Renaissance robotic system was used to plan an adequate tract for screw placement. (**b**) During the operation, the patient was placed in the prone position and stabilized under general anesthesia. The mounting equipment of the Renaissance robotic system was applied on the back, then registration with anterior–posterior and oblique plain films was performed. (**c**) Robotic assembly was performed. Subsequently, the robotic arm moved to the expected position according to the preoperative planning. The drilling execution was performed along the tract on the robotic arm. (**d**) A Kirschner wire was inserted along the tapped canal guided by the robotic system. Under radiographic plain film, the trajectory of the guided pins was identical to the expected tract, as demonstrated by the green line.

**Table 1 medicina-58-00688-t001:** Clinical characteristics and radiological results for all 19 patients.

No.	Age (Years)	Sex	T-Score	BMI	Follow-Up (Months)	Screw- Loosening	Cage Fusion
1	73	F	−2.9	27.2	87	−	+
2	57	F	−4	20.9	14	−	+
3	71	F	−4.3	23.8	59	−	+
4	62	F	−3.7	26.6	53	−	+
5	78	F	−3	25.1	55	−	+
6	63	F	−3.9	21	20	−	+
7	78	F	−3.1	31	50	−	+
8	77	F	−3.6	20.9	11.5	−	+
9	61	F	−4	28	54	−	+
10	70	M	−3.5	20.3	57	3 screws	+
11	64	F	−2.9	19.3	73	−	+
12	72	F	−3.1	39.9	13	2 screws	−
13	74	F	−3.7	35.1	40	−	+
14	62	M	−3	24.6	46	−	+
15	62	M	−3.2	23.2	12	−	+
16	81	F	−2.8	21.2	18	2 screws	−
17	68	F	−2.6	22.2	69	4 screws	+
18	70	F	−2.7	29.8	49	−	+
19	68	F	−2.6	27	39	−	+

F: female; M: male; BMI: body mass index. Screw loosening: − means no loosening screw; numbers refer to the numbers of loosened screws. Cage fusion: + means fusion; − means non-fusion.

**Table 2 medicina-58-00688-t002:** Heights of vertebral bodies evaluated preoperatively and postoperatively.

		Average (cm)		Average (cm)
Adjacent anterior 1/3	Preoperative	3.1647	Postoperative	3.1426
Adjacent middle 1/3	Preoperative	3.2058	Postoperative	3.15
Adjacent posterior 1/3	Preoperative	3.2174	Postoperative	3.2237
First fixation segment anterior 1/3	Preoperative	3.1532	Postoperative	3.1479
First fixation segment middle 1/3	Preoperative	3.1774	Postoperative	3.0005
First fixation segment posterior 1/3	Preoperative	3.1332	Postoperative	2.9395
Second fixation segment anterior 1/3	Preoperative	3.0974	Postoperative	3.0242
Second fixation segment middle 1/3	Preoperative	3.0458	Postoperative	2.9758
Second fixation segment posterior 1/3	Preoperative	2.8668	Postoperative	2.8489

This table demonstrates the average values of vertebral body height, including pre- and postoperative values. There were no significant differences in the heights of the anterior, middle, and posterior third of the vertebral body before and after surgery.

## Data Availability

All the data are available from the corresponding author upon reasonable request.

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
