# Peer review of "Prevention of Compression Fracture in Osteoporosis Patients under Minimally Invasive Trans-Foraminal Lumbar Interbody Fusion with Assistance of Bone-Mounted Robotic System in Two-Level Degenerative Lumbar Disease"

_medicina, 2022, doi:10.3390/medicina58050688_

Round 1

Reviewer 1 Report

Thank you for the opportunity to review your manuscript. I wrote some comments below.

- Introduction section
Line 55-57: "Although minimally invasive procedures have led to improvement in surgical outcomes of lumbar interbody fusion"
  Additionally, if specific details about the results of previous studies are described, it will make the surgical method of this study more interesting.

- Limitations section
There were no criteria for age in the selection of study subjects. If age can influence the study results, I think it should be included in the limitations.

- Disccusion
Please present more previous studies and compare the results of this study with those of previous studies.

Please describe the reasons for the occurrence of screw-loosening and cage non-fusion in findings of this study.

Author Response

Reviewer 1

Thank you for your valuable comments, we have addressed individual comments below and revised the text to improve the language.

  1. - Introduction section
    Line 55-57: "Although minimally invasive procedures have led to improvement in surgical outcomes of lumbar interbody fusion"
    Additionally, if specific details about the results of previous studies are described, it will make the surgical method of this study more interesting.

Ans:

Thanks for your comments. We have added some specific details about why high screw loosening rate is still observed and have rewritten the sentence as” due to repeated puncture and imperfect screw location” in line 58-59.

  1. - Limitations section
    There were no criteria for age in the selection of study subjects. If age can influence the study results, I think it should be included in the limitations.

Ans:

Thank you for raising this important point. We have added the age into our limitation sections and rewritten line 223-224 as “In addition, age should also be taken into consideration for further evaluation.”

  1. - Disccusion
    Please present more previous studies and compare the results of this study with those of previous studies.

Ans:

Thanks for your feedback. We have added some more text and references as “Finally, our results indicated that minimally invasive spinal surgery with assistance of bone-mounted robotic system is also an effective treatment to achieve spinal fixation and fusion for osteoporosis patients, compared with other methods, such as cement-augment pedicle screws or using only applied minimally invasive spinal surgery for osteoporosis patients [13,14].” (line 212-216)

  1. Please describe the reasons for the occurrence of screw-loosening and cage non-fusion in findings of this study.

Ans:

Thank you for the comment. Screw loosening and cage non-fusion cases are four cases in our study. We think the main reason could be old age and poor T-score. The reasons of these high screw loosen rate and non-fusion cage rate are due to poor fixation environment for fusion. (line 174-175)

Reviewer 2 Report

What is a two-level degenerative disease of the spine? Facet Joint degeneration? Osteochondrosis or both?

  • 19 patients are too few
  • What was the patient´s gender ?
  • T-Score should be adapted to gender and age. A T-score of -2.5 means osteoporosis but is no indication for medical treatment of osteoporosis in a female less than 75 years of age. However, a male with T-score -2.5, would only need treatment at the age of 80 years
  • Follow-up 43 months – were the patients clinically examined? What about Patient related outcome? Walking distance?

The Study concept is not clear to me. There was no randomization, patients were selected consecutively. This leads to a gender-bias. Furthermore, the cohort is only 19 patients, this is very little, concerning the large incidence of osteoporosis.

The problem of screw loosening in osteoporotic patients is well known and frequently studied. Why didn´t the authors use cement-augmented pedicle screws? This significantly reduces screw-losing in vertebras with poor bone quality and is state-of-the-art treatment in osteoporotic fracture.

Unfortunately, the article adds nothing to the current knowledge. Whether is screw is implanted by an experienced surgeon or with robotic assistance makes no differences in the literature, especially in the lumbar region. It might improve the result of implant positioning in unexperienced OR teams.

Author Response

Reviewer 2

Thank you for your valuable comments, we have replied to individual comments below. We have also enhanced our English quality with the help of Editage Language Editing Services.

  1. What is a two-level degenerative disease of the spine? Facet Joint degeneration? Osteochondrosis or both?

Ans:

Thanks for your question. We collected lumbar spondylolisthesis cases for our target cases. We also added the sentence in line 75.

  1. What was the patient´s gender ?

Ans:

Thanks for your question. The ratio of Male: Female is 3:16. (line 150-151)

  1. T-Score should be adapted to gender and age. A T-score of -2.5 means osteoporosis but is no indication for medical treatment of osteoporosis in a female less than 75 years of age. However, a male with T-score -2.5, would only need treatment at the age of 80 years

Ans:

Thank you for your comment. We know that medical treatment for osteoporosis should be adapted for gender and age. However, as surgical outcome is our main aim and goal. We focused on T-score for scientific analysis. In the future, we will keep on more strictly analysis.

  1. Follow-up 43 months – were the patients clinically examined? What about Patient related outcome? Walking distance?

Ans:

Thank you for your question. In this study, we focused on radiological results. Therefore, our mean follow-up time of 43 months attended with a last imaging examination, such as CT or MRI. We did not follow up with a clinical examination.

  1. 19 patients are too few. The Study concept is not clear to me. There was no randomization, patients were selected consecutively. This leads to a gender-bias. Furthermore, the cohort is only 19 patients, this is very little, concerning the large incidence of osteoporosis.

Ans:

Thank you for your comments and we also recognized the limitation for our study and have mentioned it in our manuscript. In the future, we hope to include a larger sample for more details.

  1. The problem of screw loosening in osteoporotic patients is well known and frequently studied. Why didn´t the authors use cement-augmented pedicle screws? This significantly reduces screw-losing in vertebras with poor bone quality and is state-of-the-art treatment in osteoporotic fracture.

   Ans:

Thank you for your valuable comments. We have mentioned that the use of cement-augmented pedicle screws is an effective method. However, cement augment still has several concerns, such as leakage. Therefore, we still hoped to offer another new treatment for dealing with osteoporosis patients. (line 212-216)

  1. Unfortunately, the article adds nothing to the current knowledge. Whether is screw is implanted by an experienced surgeon or with robotic assistance makes no differences in the literature, especially in the lumbar region. It might improve the result of implant positioning in unexperienced OR teams.

Ans:

Thank you for your comments and we will try our best to offer effective and safe treatment for osteoporosis patients to achieve the same result as experienced OR teams.

Round 2

Reviewer 2 Report

Dear Authors, 

thank you for your effort to improve the manuscript. The revisied version is laudable and I recommend publication. 

Best regards
